# From Cell Culture to Organoids-Model Systems for Investigating Prion Strain Characteristics

**DOI:** 10.3390/biom11010106

**Published:** 2021-01-14

**Authors:** Hailey Pineau, Valerie L. Sim

**Affiliations:** 1Department of Medicine, University of Alberta, Edmonton, AB T6G 2B7, Canada; pineau@ualberta.ca; 2Centre for Prions and Protein Folding Diseases, University of Alberta, Edmonton, AB T6G 2R3, Canada

**Keywords:** prion, cell culture, organotypic slice culture, organoids, stem cell, strains

## Abstract

Prion diseases are the hallmark protein folding neurodegenerative disease. Their transmissible nature has allowed for the development of many different cellular models of disease where prion propagation and sometimes pathology can be induced. This review examines the range of simple cell cultures to more complex neurospheres, organoid, and organotypic slice cultures that have been used to study prion disease pathogenesis and to test therapeutics. We highlight the advantages and disadvantages of each system, giving special consideration to the importance of strains when choosing a model and when interpreting results, as not all systems propagate all strains, and in some cases, the technique used, or treatment applied, can alter the very strain properties being studied.

## 1. Introduction

Transmissible spongiform encephalopathies (TSEs) are a group of invariably fatal, rapidly progressive neurodegenerative diseases for which there are no cures. These diseases are caused by the misfolding of the prion protein (PrP), which exists ubiquitously in the brain and other tissues [1,2]. In its normal alpha-helical state, PrP^C^ is non-toxic and has many proposed functions [3,4]. In TSEs, PrP adopts a beta sheet-rich conformation (denoted PrP^Sc^) and forms fibrils and aggregates that are often cytotoxic [5,6]. Because PrP^Sc^ templates the conversion of PrP^C^ to this aberrant state, TSEs are infectious and spread at both the cell and host levels [7].

A number of mammals can be affected by TSEs, including scrapie in sheep and goats, chronic wasting disease (CWD) in cervids, and bovine spongiform encephalopathy (BSE) in cattle [2,6]. In humans, the most common TSE is Creutzfeldt–Jakob disease (CJD), which can arise sporadically (denoted sCJD), iatrogenically through transmission of PrP^Sc^ from contaminated surgical equipment or dura mater grafts, or genetically due to mutations in the *Prnp* gene, which encodes PrP [8]. Additionally, variant CJD (vCJD) is a newer form of the disease that arose from ingestion of BSE prions [9]. Other genetic TSEs in humans include Gerstmann–Sträussler–Scheinker syndrome and fatal familial insomnia [10,11]. Kuru, another human TSE, was endemic among the Fore people of Papua New Guinea and was transmitted via cannibalistic funeral rituals [12,13].

There are many different model systems in use for investigating the mechanisms of prion disease and screening drugs, from simple cell culture to organoids. However, many of the drugs shown to be effective in cell culture fail to show efficacy in vivo or in human patients [14]. This is likely due in part to the vast heterogeneity of prions, which exist as different strains. Strains are defined clinically by differences in incubation period and lesion profiles and are thought to arise from different conformations of PrP^Sc^ [15,16]. Different strains can also be characterized biochemically by variation in PrP^Sc^ glycosylation pattern, electrophoretic mobility following PK-digestion, and conformational stability [17].

In humans, sCJD exists as six different strains that are classified based on the *Prnp* genotype at codon 129 (either methionine or valine), and the molecular weight of the protease-resistant core of PrP^Sc^ (21kDa in type 1 or 19kDa in type 2). These six strains are therefore denoted MM1, MM2, MV1, MV2, VV1, and VV2, with each having unique disease characteristics [18].

Additionally, several strains of prion disease have emerged in a laboratory setting after the serial passaging of scrapie, BSE, or CJD isolates in laboratory animals of particular genetic backgrounds. Typically, the incubation period becomes shorter with subsequent passages and disease characteristics drift and stabilize, giving rise to mouse, hamster, or rat-adapted strains [17]. See Table 1 for an overview of the different species and strains of prion disease that are covered in this review.

The efficiency or ability of strains to propagate depends on the culture system used, with cells of a particular type and species only able to propagate a subset of strains [19]. Moreover, different strains have been shown to have vastly different responses to drugs [20,21]. Therefore, it is paramount that model systems are able to recapitulate strain features seen in vivo. Of course, this must be balanced with considerations of time, cost, and ethics. In this paper, we discuss and evaluate the current models available for studying prion disease with particular focus on how these models have enhanced our understanding of strains.

## 2. Immortalized Cell Lines

The first in vitro systems used to study prion propagation were immortalized cell lines. While quick, cost-effective, and easy to maintain, these culture systems must be continuously passaged, meaning that prions must accumulate at a rate faster than cells divide for stable infection to occur. Moreover, the continuous passage of cells does not necessarily reflect the in vivo situation, in which differentiated, non-dividing neurons are infected and continue to accumulate PrP^Sc^ for months or years as the disease progresses. Additionally, many of these cell lines are derived from tumors and are, therefore, unstable and subject to genetic drift over time. It is also important to note that each cell line is generally permissive to infection with only a subset of the strains that propagate in vivo (summarized in Table 2). Nonetheless, much has been learned about prions from these systems, including the discovery of cofactors necessary for prion propagation, the effects of glycosylation on PrP^Sc^ infectivity, mechanisms of intercellular spread, and the uncovering of many potentially useful drug candidates.

### 2.1. Neuron-Like Cell Lines

#### 2.1.1. Scrapie Mouse Brain (SMB) Cells

Among the first cell lines to be used in prion research was the scrapie mouse brain (SMB) line, which was developed by Clarke and Haig from the brain of a mouse infected with the Chandler strain of mouse-adapted scrapie. SMB cells, which were found to be of mesodermal origin, remained persistently infected after several passages as evidenced by their ability to induce typical scrapie when inoculated into mice [22]. It was subsequently found that these cells could be cured with pentosan sulfate and that they could propagate several other strains of mouse-adapted scrapie, including 22F, 139A, and 79A. The 263K hamster-adapted scrapie strain, which is often used to test the scrapie species barrier, was not able to propagate in this system [23].

#### 2.1.2. PC12 Cells

Another early cell line used in prion research was the PC12 rat pheochromocytoma line. Upon exposure to nerve growth factor (NGF), these cells differentiate and acquire neuronal properties, including the synthesis of neurotransmitters [24]. The cells have been shown capable of propagating the 139A and ME7 strains of mouse-adapted scrapie, but not 263K hamster adapted scrapie [24,25]. Interestingly, although these are rat cells, they were unable to propagate the 139R rat-adapted scrapie strain [25].

#### 2.1.3. N2a and Other Neuroblastoma Cells

N2a cells, a line of mouse neuroblastoma cells, are among the most extensively used in prion research. These cells are very heterogenous, with subclones expressing varying levels of PrP^C^ [65]. Moreover, different subclones are able to propagate different prion strains to different extents, which does not seem to be related to PrP^C^ expression. For example, the PK1 and R33 subclones express similar levels of PrP^C^, but while PK1 is highly susceptible to both 22L and RML infection, the R33 subclone can only propagate 22L [28]. Along with the LD9 line of fibroblasts and CAD5 catecholaminergic cells, the PK1 and R33 N2a subclones are used in the cell panel assay for strain typing of mouse adapted prions [29]. Additionally, PK1 cells are used in the scrapie cell assay, which is a widely used technique for measuring titers of prion infectivity [28]. These techniques are equally as sensitive as mouse bioassay, but much faster and less expensive. However, the use of the scrapie cell assay is limited to strains that are able to propagate in N2a cells, which include 22L, RML, 139A and 79A, but not the ME7, 22A, 263K or the 301C strain of mouse-adapted BSE [28,66]. Recently, however, with several rounds of subcloning, the PME2 line of N2a cells was developed, which was able to propagate ME7 prions [31]. Other neuroblastoma cell lines have also been shown to propagate scrapie prions, including the DK, DL, and C-1300 lines, which propagate the RML but not 263K strain [67]. Additionally, SN56 cells are a mouse septal cholinergic neuroblastoma line, which have been shown to propagate the 22L, RML, and ME7 strains, but not 87V or 263K [34]. The SH-SY5Y human neuroblastoma line was also reported to propagate human CJD prions [68], however this result is widely disputed and has never been repeated.

#### 2.1.4. CAD5 Cells

CAD5 cells are a line of mouse catecholaminergic cells, which have also been used extensively in prion research. They are susceptible to a wider array of strains than N2a cells. In addition to RML, 22L, 139A and 79A, they are able to propagate ME7 and the 301C mouse-adapted BSE strain. They are unable to propagate the hamster-adapted 263K strain, showing that the mouse-hamster species barrier is intact [29,30]. Because of their wide strain permissibility, a couple of groups have used CRISPR to knock out mouse PrP in CAD5 cells and reconstitute them with PrP of different species. Walia et al., found that CAD5 cells expressing various genotypes of cervid PrP could propagate both mule deer (MD) and white-tailed deer (WTD) CWD prions. CAD5 cells expressing bank vole PrP (bvCAD5 cells) were also able to propagate these strains of CWD, but with lower efficiency. These bvCAD5 cells could also be infected with 22L prions [33]. Additionally, Bourkas et al., demonstrated that CAD5 cells expressing hamster PrP could propagate the 263K, hyper (HY) and 139H strains of hamster-adapted prions, but not the drowsy (DY) strain [32]. Thus, CRISPR has proven to be a useful tool for investigating species barriers. Specifically, we can explore whether the expression of species-specific PrP is sufficient for infection, or whether other conditions, such as species-specific cofactors, are also required.

#### 2.1.5. GT1 Cells

The GT1 cell line is a murine hypothalamic cell line, which is able to propagate Chandler, RML and 22L mouse-adapted scrapie prions, natural Kanagawa scrapie, and the Fukuoka-1 (Fu-1) and SY strains of mouse-adapted CJD prions [26,35,36,37,69]. While many authors have reported a lack of cytotoxic effects in most prion-infected cell lines, Schatzl et al., demonstrated that RML-infected GT1 cells exhibited changes in morphology, including the clumping of chromatin, nuclear pyknosis, the presence of autophagic vacuoles, and the formations of abnormal non-monolayer cell clusters. When GT1 cells were transfected to express the trkA receptor, which binds nerve growth factor (NGF), the addition of NGF was shown to prevent these morphological changes [35].

#### 2.1.6. CRBL Cells

The CRBL line was developed from the cerebellum of a transgenic mouse lacking expression of p53, a protein involved in cell cycle arrest. These cells were found to be heterogenous, with some resembling neurons, others resembling astrocytes, and some fibroblasts. The karyotype of these cells was also quite variable, ranging from 66 to 78 chromosomes. Although 139A and RML are both strains derived from Chandler prions, these cells were able to propagate the 139A strain and not RML [38].

#### 2.1.7. 1C11 Cells

1C11 cells are a murine cell line that behave like neuronal stem cells. Nondifferentiated cells can propagate Chandler, 22L, and Fu-1, with Fu-1 producing the greatest amount of PrP^Sc^; 22A and ME7 fail to propagate in 1C11 cells [39]. These cells undergo differentiation into serotonergic neurons with addition of dibutyrylcyclic AMP or differentiation into noradrenergic neurons when both dibutyrylcyclic AMP and DMSO are added. Therefore, they are useful for investigating the effects of prion infection on neurotransmitter dynamics. Mouillet-Richard et al., found that the serotonergic phenotype of differentiated cells produced less 5-HT and had greater 5-HT catabolism when infected with Fu-1. Prion infection also led to the accumulation of 5-HT oxidation products in these cells [39].

#### 2.1.8. Neural Cell Lines from Prnp^0/0^ Mice

HpL3-4 is an immortal hippocampal line derived from *Prnp*^0/0^ mice. Maas et al., used retroviral transduction to induce expression of murine PrP with the 3F4 epitope, which enabled distinction between original inoculum (lacking the 3F4 epitope) and de novo PrP^Sc^. These cells were shown to propagate 22L prions with a greater percentage of cells becoming infected with each passage. Then, the authors produced cells with various *Prnp* mutations at sheep-specific residues, some of which were found to impair propagation of 22L [40]. This experiment highlights the usefulness of *Prnp*^0/0^ cell lines for investigating the effects of *Prnp* mutations on species barriers.

Other experiments have made use of *Prnp*^0/0^ neural cell lines to investigate the necessity of the GPI anchor for prion infection. CF10 cells are another line of neural cells derived from *Prnp*^0/0^ mice. In an experiment conducted by McNally et al., these cells were engineered to express either normal GPI-anchored mouse PrP, anchorless PrP, or both. Only cells that had GPI-anchored PrP could propagate 22L prions [41]. In a similar experiment, hippocampal NpL2 cells (also derived from *Prnp*^0/0^ mice) were engineered to express either wildtype PrP or PrP with a flexible linker instead of a GPI anchor signal sequence, which led to PrP being inserted into non-raft regions of the plasma membrane. Cells expressing normal GPI-anchored PrP could be infected with RML and 22L prions, but cells expressing non-GPI anchored PrP could not be infected by either strain. Therefore, even when PrP was localized to the membrane, a GPI anchor was necessary for infection [70].

### 2.2. Microglial Cell Lines

It is generally accepted that microglia are protective in prion disease [71,72]. However, microglia have been shown to contribute to pathology in other protein misfolding neurodegenerative diseases, such as through the release of inflammatory cytokines or reactive oxygen species (ROS) [73]. Much remains to be learned about the role of these cells in prion disease. Iwamaru et al., generated two lines of immortalized microglia; the MG6 line from c57 mice, and the MG20 line from Tga20 mice, which overexpress PrP. MG20 cells were able to propagate Chandler, ME7, natural Obihiro scrapie, and a BSE agent. However, MG6 cells, which express 9× less PrP^C^, were not able to propagate any of these strains [42]. Munoz-Gutierrez et al., developed an immortalized line of ovine microglia through transfection of primary microglia culture with human telomerase reverse transcriptase. This prevented cell senescence without increasing the rate of cell division. These hTERT ovine microglia were susceptible to infection by several natural scrapie isolates, including some that did not propagate in Rov cells (described later). Interestingly, one of the microglial sublines was capable of forming multilayers and therefore did not require a media change for 4 weeks. This subline was able to propagate additional scrapie isolates [43]. Therefore, the selection of clones or sublines that can be sustained longer between media changes is one strategy for enhancing the range of susceptibility to more slowly-propagating prion strains.

### 2.3. Astrocyte Cell Line: C8D1A

Recently, Tahir et al., demonstrated that C8D1A cells, an immortalized astrocyte line, are susceptible to 22L and RML prions, but not ME7. Interestingly, differences were observed between 22L- and RML-infected cells. Confocal imaging revealed PrP puncta in 22L-infected C8D1A cells, but a much weaker, diffuse staining in RML-infected cells. Additionally, Western blot showed much lower levels of PK-resistant PrP^Sc^ in cells infected with RML as compared with 22L. However, real-time quaking-induced conversion (RT-QuIC) demonstrated that RML-infected cells had greater seeding activity than 22L-infected cells [44].

### 2.4. Schwann Cell Lines: MovS6 and MovS2

Archer et al., developed an immortalized cell line from the dorsal root ganglia of tg301 mice, which express ovine PrP. These cells were discovered to be Schwann-like in nature. Two subclones, MovS2 and MovS6, were found to express 6x and 8x physiological PrP levels respectively, and could propagate PrP^Sc^ from a natural 127S scrapie strain that was first passaged in TgOv mice [45]. Neale et al., further showed that MovS6 cells were susceptible to several natural scrapie cases from sheep with the VRQ allele (i.e., the allele associated with the greatest scrapie infectibility in sheep) [46]. Schwann cell lines may be particularly useful for investigating the propagation of PrP^Sc^ in the peripheral nervous system and neuroinvasion. When Archer et al., examined TgOv mice that had been peritoneally infected with scrapie, they found that Schwann cells and satellite cells often contained PrP^Sc^ and were usually adjacent to a neuron also containing PrP^Sc^ [45].

### 2.5. Fibroblast Cell Lines

Vorberg et al., found that two lines of mouse fibroblasts, NIH/3T3 and L929, were both able to accumulate and propagate PrP^Sc^ after exposure to 22L brain homogenate. L929 was shown to be more susceptible, with 47% of clones accumulating detectable levels of PrP^Sc^ compared to 12% of NIH/3T3 cells [48]. It has also been demonstrated that L929 cells can propagate the RML, ME7, 139A, and 79A strains, but not 87V or 301C [29,30,48]. LD9, a highly susceptible subclone of L929 cells, is used in the cell panel assay for prion strain typing [29]. L929 cells have also been used to elucidate differences in endocytic routes for different strains; Fehlinger et al., found that when clathrin heavy chain was knocked out in L929 cells, infection with RML prions was impaired, while infection with 22L was not [74].

Recently, Walia et al., used CRISPR to knock out mouse PrP in the MEF strain of murine fibroblasts. The cells were reconstituted with cervid PrP of the wildtype, 116G, or 138N genotypes. In all cases MEF cells were shown capable of propagating both MD and WTD CWD. MEF cells expressing bovine PrP were also infectible with both types of CWD, but accumulated lower levels of PrP^Sc^ [33]. Thus, in combination with genetic engineering, fibroblast cell lines may be useful for investigating species barriers and species-specific mutations.

### 2.6. RK13 Cells

RK13 cells are rabbit kidney epithelial cells that express negligible levels of rabbit PrP [53]. Over the past several years, they have been engineered to express PrP of many different species and have therefore facilitated investigation of strains and species barriers. Courageot et al., developed moRK13 cells, which express murine PrP and can propagate Chandler, 22L, and Fu-1 prions, but not ME7 [50]. Recently, Wusten et al., engineered RK13 cells to express mouse PrP fused with either half of a split Gaussia luciferase. When PrP dimerizes in these cells, a luminescent signal is produced. Dimerization of PrP was detected in these cells upon incubation with several different strains of scrapie, including RML, 22L, ME7, 79A, 87V, 22A, and 263K [75].

Rov cells have also been generated by transfecting RK13 cells to express ovine PrP with the VRQ genotype. Through this method, Vilette et al., produced the Rov9 line, which expresses levels of ovine PrP comparable to that of the sheep brain. They found that these cells were able to propagate PrP^Sc^ for several passages when incubated with PG127 and LA404 natural scrapie isolates, both containing the VRQ genotype [53]. Arellano-Anaya et al., also showed that RK13 cells expressing ovine PrP could be infected with 127S scrapie [51]. Sabuncu et al., produced RK13 cells expressing the ARR allele, which is known to provide resistance to scrapie in vivo [76]. In this case, the RK13 cells did not develop detectable levels of PrP^Sc^ when incubated with the VRQ scrapie isolates [54]. Likewise, Neale et al., found that Rov9 cells expressing the VRQ allele could only propagate VRQ scrapie prions and failed to propagate natural cases with other genotypes [46].

Like sheep, goats are also naturally affected by scrapie. In goats, two genotypes of the *Prnp* gene (haplotype 1 and 2) are both considered wildtype, with haplotype 2 being identical to the ovine wildtype ARQ allele. Haplotype 1 is identical to haplotype 2, except for a substitution from serine to proline at codon 240. Goats with either of these haplotypes are very susceptible to scrapie, while other haplotypes confer more resistance [77]. Dassanayake et al., demonstrated that RK13 cells engineered to express caprine PrP with haplotype 2 (cpRK13 cells) could be infected with natural caprine scrapie of haplotype 1,1 or haplotype 1,2, but not to natural cases with haplotypes 3 or 4. However, Tg338 mice, which express caprine scrapie with haplotype 2, were able to propagate these natural haplotype 3 and 4 isolates [56]. Therefore, these mice might possess specific cofactors necessary for propagation that are lacking in RK13 cells. Subsequently, they found that after 2 serial passages of haplotype 3 or 4 scrapie isolates in Tg338 mice, cpRK13 cells were able to propagate these mouse-adapted isolates [56].

RK13 cells have also been engineered to express cervid PrP. Interestingly, Bian et al., showed that RK13 cell expressing both elk PrP and the HIV-1 GAG precursor protein (RKE-Gag cells) were able to propagate PrP^Sc^ for multiple passages when exposed to either deer or elk CWD prions. However, RKE cells without the HIV-1 GAG precursor protein could not propagate PrP^Sc^ [57]. This fits the findings of other groups that have shown that retroviral infection or the expression of retroviral proteins such as the Gag polyprotein enhances prion infection and release of PrP^Sc^ in cell culture [78]. It has been proposed that retroviral proteins, which localize to the same detergent-resistant microdomains as PrP, may be important cofactors in prion disease [78]. Consequently, the expression of such proteins could be a useful strategy for expanding the range of strains and species that cell lines are susceptible to.

RK13 cells expressing human MM PrP (huRK13 cells) have also been generated. However, no PrP^Sc^ was detected in huRK13 cells exposed to MM2 CJD brain homogenate or to the mouse-adapted M1000 CJD strain. It was found that moRK13 cells could propagate M1000 but not the MM2 CJD brain homogenate. However, after serially passaging the MM2 CJD brain homogenate in mice, the moRK13 cells were able to propagate the resultant mouse-adapted MU-02 prions [52]. Therefore, RK13 cells have proven to be a versatile cell line for investigating strain adaptation and species barriers.

### 2.7. MDBK Cells

The immortalized Madin-Darby Bovine Kidney (MDBK) cell line has been studied for its potential to support infection with BSE prions. Oelschlegel et al., found that although MDBK cells were able to propagate PrP^Sc^ upon exposure to several natural ovine scrapie isolates (including those with the ARQ allele), they were not infectible upon exposure to any natural or mouse-adapted BSE agents [58]. However, Tark et al., used a lentiviral expression system to increase expression of bovine PrP in MDBK cells and reported that they were capable of propagating natural BSE prions, but only for 20 passages. After subcloning, however, they discovered a clone (denoted M2B) that was able to sustain infection for more than 83 passages [59]. Consequently, this study illustrates that for some cell lines, increasing expression of PrP is a practical method for enhancing infectibility.

### 2.8. C2C12 Myotubes

Because skeletal muscle is known to accumulate PrP^Sc^ and pathology in many prion diseases of both animal and humans [79,80,81], Dlakic et al., investigated whether C2C12 murine myoblasts could be infected with and propagate scrapie. C2C12 myoblasts are non-differentiated mononuclated fibroblast-like cells that divide every 14–16 h. However, once they reach confluence, they fuse together and differentiate to form multinucleated myotubes that are no longer capable of cell division [81]. When Dlakic et al., exposed C2C12 cultures to 22L brain homogenate, no PrP^Sc^ was detected after 8–10 passages. When zeocin-resistant C2C12 cells were cocultured with 22L-infected N2a cells, however, C2C12 cells were found to propagate PrP^Sc^ for at least 4 serial passages after selectively killing N2a cells with zeocin. Intriguingly, C2C12 cells could not be infected by 22L-N2a cells when the two types of cells were separated by a PET membrane. SMB cells were also ineffective at inducing infection in C2C12 cultures, which suggests that direct contact with an infected neuron-like cell (such as an N2a cell) might be necessary for C2C12 cells to accumulate PrP^Sc^ [81].

Herbst and colleagues also investigated the potential for C2C12 cultures to accumulate PrP^Sc^. As in the previous study, they found that non-differentiated myoblasts were incapable of accumulating PrP^Sc^ upon exposure to scrapie brain homogenates. However, when cultures were deprived of serum to induce differentiation into myotubes, sustainable infection was achieved with RML, 22L, and ME7 prions. The hyper strain of hamster-adapted mink encephalopathy failed to propagate, demonstrating the integrity of the species barrier. Of the three permissive scrapie strains, RML led to the greatest accumulation of PrP^Sc^ [60]. The permissiveness of myotubes to prion infection may be due to their four to six times greater PrP^C^ expression than their non-differentiated myoblast counterparts [81]. Moreover, since differentiated myotubes no longer divide and do not require passaging, they can be cultured for a longer time allowing them to accumulate more PrP^Sc^ [60]. The use of long-term non-dividing cell cultures could be of great use in prion research as this more closely resembles the in vivo situation in which non-dividing neurons are chronically infected for months to years.

### 2.9. Cells of the Lymphoid Reticular System

Because vCJD is thought to spread from the periphery to the central nervous system via the lymphoid reticular system [82], the ability to propagate prions in such cell types is of great importance. For this purpose, Akimov et al., prepared primary culture from the spleen of an SJL/OlaHsd (ola) mouse. Like the related SJL mice, Ola mice are known to develop tumors of the lymph nodes, Peyer’s patches, and spleen in old age [83,84]. They found that the cells became immortalized by 8 weeks and subsequently transformed. These cells (denoted tSP-SC cells) had high levels of PrP^C^ expression as well as CD13, which is a marker of macrophages, monocytes, mast cells and immature dendritic cells. They also expressed CD44 glycoprotein, which is a marker of stromal cells. Upon exposure to either Fu-1 or a mouse-adapted vCJD strain, the tSP-SC cells were able to stably propagate PrP^Sc^ for over 30 passages. Notably, strain differences were observed, with PK-resistant PrP^Sc^ having a lower molecular weight in vCJD-infected cells compared to those infected with Fu-1 [61]. In another experiment from the same group, an immortal cell line was derived from the bone marrow of an Ola mouse. These cells had characteristics of adipocytes, and once transformed, could stably propagate Fu-1 prions for 57 passages [62]. In an additional study, two separate immortalized cell lines (O1BM and O2BM) were prepared from the bone marrow of an Ola mouse. O1BM could be infected with Fu-1 brain homogenate, but O2BM could not be. Further investigation revealed that the O2BM cell line had a higher proportion of cells positive for stem cell antigen 1 and that these cells differentiated much more efficiently [63]. Such results are in agreement with many other experiments that show that differentiated cells are much better able to sustain PrP^Sc^ infection than their stem cell counterparts [60,85].

In addition to mouse cells from the lymphoid reticular system, Krejciova et al., cultured cells of the human HK line, which have features of follicular dendritic cells. The cells, which express the more resistant VV genotype of human PrP, were exposed to MM vCJD brain homogenate. Although immunostaining showed a punctate pattern of PrP^Sc^ within lysosomal/endosomal compartments of the cells immediately after exposure, they failed to propagate PrP^Sc^ over multiple passages [64]. Perhaps such cells with the more susceptible MM phenotype would be better able to sustain infection. Nonetheless, culturing cells of the lymphoid reticular system may be useful for investigating uptake of PrP^Sc^ and spread to other cell types.

## 3. Primary Culture

To overcome the limitations of immortalized cell cultures, many researchers have turned to primary cultures to study prion infection. Unlike the immortalized cell culture described so far, primary cultures can be prepared from the actual cell types of the nervous system that are most affected by prion disease, including neurons, astrocytes, and microglia. Furthermore, these cells are post-mitotic, meaning that they do not divide, and therefore do not need to be continuously passaged during culturing. For this reason, PrP^Sc^ has more time to accumulate in primary cultures and the downstream effects of such accumulation can be elucidated. However, because primary cultures are difficult to maintain due to a base level of apoptosis, they can only be cultured for so long. Additionally, since serial passaging cannot be used to remove residual inoculum, primary culture experiments must always be done in parallel with equivalent *Prnp*-knockout cultures to ensure that de novo PrP^Sc^ is being produced. Despite such limitations, primary culture has proven useful for studying strain-specific cytotoxic effects, which often cannot be observed in immortalized cell culture systems. Table 3 summarizes the strains that are able to propagate in different types of primary cell culture systems.

Several experiments have been done to investigate scrapie infection of cerebellar granular neurons. Cronier et al., prepared such cultures from tg338 mice, which express ovine PrP. These cultures were found to contain greater than 95% neurons, 4–5% astrocytes, and less than 1% microglia. When exposed to either cell lysates from scrapie-infected Mov cells or to brain homogenate from Tg338 mice infected with natural scrapie, PK-resistant PrP^Sc^ became detectable in the cultures between 14 and 21 days and continued to accumulate. Additionally, immunostaining revealed a punctate pattern of PrP^Sc^ deposition in both infected neurons and astrocytes, which was dependent on treatment with GdnSCN. Because it is very hard to maintain pure primary neuronal cultures for extended periods of time, the authors also prepared a culture of highly enriched neurons on top of a supportive layer of astrocytes from a *Prnp*-knockout mouse. The neurons in this culture were also susceptible to infection, showing that the presence of PrP-expressing astrocytes isn’t necessary. Moreover, while cytotoxic effects are rarely observed in immortalized cell lines, there was a two times increase in the rate of apoptosis by 28 days post infection in these primary cultured CGNs compared to non-infected controls [89].

Prions of other species can also be propagated in CGN primary cultures. The ovine-PrP-expressing cultures prepared from Tg338 mice were shown to be capable of propagating 139A prions [89]. Additionally, primary CGN culture has also been prepared from Tga20 mice, which overexpress mouse PrP^C^, and were found to be susceptible to 139A, 22L, ME7, and Fu-1 prions. Of these strains, ME7 produced the least PrP^Sc^. CGN cultures from tg7 mice, expressing hamster PrP, could propagate the Sc237 subclone of 263K and the hamster-adapted 139H strain. Excitingly, primary CGNs prepared from MM human PrP-expressing tg650 were susceptible to infection with human type 1 CJD inoculum, with PK-resistant PrP^Sc^ being detected by 28 days after exposure [86]. Similarly, Hannaoui et al., found that CGN cultures expressing MM PrP were susceptible to both sporadic and iatrogenic MM1 CJD isolates as well as an MM vCJD isolate. CGN cultures expressing VV PrP were also found to be susceptible to a VV2 sCJD isolate [21] Therefore, unlike immortalized cell cultures, primary culture is a promising tool for studying the propagation of non-mouse adapted CJD agents.

Primary culture has also been prepared from neurons of other types and brain regions, which has been particularly useful for studying strain tropism. In one experiment, primary cultures were prepared from either CGNs, striatal neurons, or cortical neurons of c57 mice and exposed to either 139A, ME7, or 22L prions. All strains could be propagated in CGNs, but 22L led to the fastest accumulation of PrP^Sc^, and ME7 was the slowest. Furthermore, the greatest cell death was observed in 22L-infected CGNs, followed by ME7, then 139A. However, in striatal and cortical neuronal cultures, ME7 failed to propagate. In striatal cultures, 139A accumulated faster than 22L and led to greater cytotoxicity. In cortical cultures, the kinetics and cytotoxicity of 22L and 139A were very similar [87]. These results reflect the in vivo situation, in which 22L is known to target the cerebellum and 139A is known to target the striatum [93]. Additionally, Fang et al., have demonstrated that primary hippocampal neurons from c57 mice are susceptible to infection with RML prions. Infection led to a decrease in the density of dendritic spines compared to PrP-knockout cultures exposed to RML [91].

Primary culture has enhanced our understanding of how prion strains interact with cofactors. Hasebe and colleagues also showed that different strains of mouse-adapted scrapie interact uniquely with different compliment factors in primary cortical mouse neurons. For Chandler-infected neurons, exposure to complement factors C1q, C3 and C9 led to an increase in membrane permeability and a decrease in levels of PrP^Sc^. However, in 22L-infected neurons, only C3 was associated with an increase in membrane permeability, and although levels of PrP^Sc^ initially declined, there was a subsequent increase [90].

Cocultures of primary cells have facilitated our understanding of how PrP^Sc^ is transferred between cell types. In an experiment conducted by Victoria et al., 22L-infected primary astrocytes were added to non-infected neurons. Subsequently, PrP^Sc^ was only detected in neurons in which there was cell-to-cell contact between an astrocyte and a neuron. However, cell-to-cell contact was not necessary for propagation of PrP^Sc^ between astrocytes as conditioned media from infected astrocytes could be used to induce infection in naïve astrocytes. Nevertheless, cell-to-cell contact was found to result in more efficient trafficking of PrP^Sc^ between astrocytes [88].

## 4. Stem Cell-Derived Cultures

Induced pluripotent stem cells (iPSCs) are emerging as an invaluable tool in prion research. Because they can be generated from any individual organism, the genotypes of the resultant cultures are not restricted to available transgenic animals, meaning that the relationship between prion infection and genetic background can be investigated on a much larger scale. Moreover, iPSCs allow for the study of prion infection and propagation in primary human cultures, which due to ethical reasons, was previously unfeasible. This is an important step forward since many drug candidates that were shown to be promising in animals failed to be effective in clinical trials.

Recently, Krejciova et al., produced human iPSC-derived astrocytes, which were exposed to various CJD innocula. Astrocyte precursor cells were never able to propagate any CJD strain. Astrocytes expressing MM PrP were able to propagate MM vCJD, MM1 sCJD and VV2 sCJD, with the vCJD prions propagating the most efficiently. Astrocytes expressing MV PrP were also able to propagate vCJD, but PrP^Sc^ took much longer to accumulate. In contrast, astrocytes that expressed VV PrP were unable to propagate MM1 sCJD or efficiently propagate vCJD. However, these VV astrocytes accumulated an abundance of PrP^Sc^ after exposure to VV2 CJD, though they were unable to propagate VV1 CJD [85]. This experiment illustrates the practicality of iPSC-derived cultures for untangling the relationship between genotype and strains in human prion disease.

It has also been possible to generate iPSC-derived cultures from patients with specific *Prnp* mutations, which are known to result in familial prion disease. Such cultures open the possibility for personalized drug-screening for individual patients. In one such example, iPSCs were produced from dermal fibroblasts harvested from a GSS patient with the Y218N mutation. After differentiating these cells into neurons, they were monitored for signs of prion pathology. Although no PK-resistant PrP^Sc^ was detected during the culturing period of 120+ days, other pathological features, including phosphorylation of tau, astrogliosis, chromatin condensation, and increased apoptosis were observed relative to control. However, when the authors attempted to infect these cultures with sCJD or Y218N GSS prions, no PrP^Sc^ was detected beyond 2 weeks. Therefore, the cells could not be stably infected [94].

## 5. Neurospheres

A major limitation of the model systems discussed so far is that they only contain one or two cell types and lack the intricate connections between cells found in the in vivo brain. One system that overcomes some of these limitations is neurospheres, which are aggregates of stem cells isolated from embryonic mouse brains and grown in suspension. Neurospheres are heterogenous balls of tissue containing both non-differentiated neuroprogenitor cells and differentiated neurons and astrocytes [95]. They also have the advantage of being self-renewing and can therefore be repeatedly passaged, overcoming the proliferation limitations of primary culture [19].

Giri et al., were the first to demonstrate prion infection of neurospheres. The neuropsheres were prepared from embryonic FVB mice, PrP-overexpressing tg4053 mice, and from *Prnp* knockout mice and exposed to RML prions for 4 days. 24 days after exposure, PrP^Sc^ was no longer detected in the knockout neurospheres, but low levels were observed in both FVB and tg4053 neurospheres and continued to accumulate, with much more present in tg4053 than FVB neurospheres by 36 days. Immunostaining revealed PrP-positive puncta in RML-infected neurospheres but not in non-infected controls [96]. Other authors have reported that murine neurospheres are able to propagate several other prion strains, including Chandler, 22L, ME7, and mouse-adapted BSE and GSS [97,98]. However, sustained infection depended on the differentiation of the neurospheres, which could be achieved by removal of growth factors from the media. As expected from previous studies with stem cells, neurospheres were found to express higher levels of PrP once differentiated [97]. Recently, neurospheres from tg5037 mice expressing elk PrP were shown to be capable of propagating PrP^Sc^ from both elk and deer CWD inoculum [99]. Neurospheres are therefore infectable with a wide array of prions of different strains and species.

Herva et al., have demonstrated that neurospheres may be useful for studying strain differences. In their experiment, differentiated murine neurospheres were exposed to RML, 22L, or 301C brain homogenate. Although other have shown the ability for murine neurospheres to propagate mouse-adapted BSE strains [98], 301C failed to propagate in this case. However, 22L and RML both led to sustained infection. Different kinetics were observed between these strains, with 22L-infected Tga20 neurospheres accumulating detectable levels of PrP^Sc^ before RML-infected neurospheres. Unique kinetic profiles were also observed for infected neurospheres derived from mice of different genetic backgrounds, suggesting that neurospheres could also facilitate study of the interaction between strains and host genotype [97].

The potential usefulness of neurospheres for screening drugs has also been illustrated. As with primary cultures, neurospheres infected with Chandler, ME7, or 22L prions exhibit cytopathic changes, including increased membrane permeability, death of astrocytes, and detachment of from the coverslips that they were grown on. The use of a pan-caspase inhibitor was effective in mitigating these changes [99].

## 6. Brain Aggregates

Brain aggregates (BrnAggs) are another potential model system for studying prion disease and have many similarities to neurospheres. BrnAggs are prepared from whole fetal mouse brains, which are cultured in constantly rotating flasks. After about two weeks of culture, they form spheres of about 1 mm in diameter with neurons, astrocytes, microglia, and oligodendrocytes. The neurons in BrnAggs are mature with axons, dendrites, and synaptic connections [100]. Thus, BrnAggs possess some of the cytoarchitecture present in vivo. Bajsarowicz et al., demonstrated that BrnAggs from FVB mice could be infected with RML prions. Levels of PrP^Sc^ accumulated throughout the 35-day culturing period, and cytopathic changes were observed, including the loss of dendritic spines, activation of microglia, and formation of autophagic vacuoles. Like neurospheres, BrnAggs could be a useful tool for screening anti-prion therapeutics. In this system, a gamma secretase inhibitor was found to prevent loss of dendritic spines, and quinacrine and rapamycin were both found to be effective at reducing levels of PrP^Sc^ [100].

## 7. Organotypic Slice Culture

Organotypic slice culture is an ex vivo system in which mouse brain slices are maintained for weeks to months. Many different brain regions can be cultured, including cerebellum, hippocampus, and cerebral whole brain slices. Organotypic slice culture has been used extensively to study protein folding diseases, including prion disease [101]. Falsig and Aguzzi developed the Prion Organotypic Slice Culture Assay (POSCA) in which 350 μm-thick cerebellar brain slices from mouse pups are exposed to infectious prions and then cultured for several weeks and accumulate PrP^Sc^ [102]. POSCA provides a valuable intermediate between in vitro and in vivo systems. Unlike cell cultures, slice cultures contain many relevant cell types with in vivo-like cytoarchitecture preserved. In contrast to live animal models, slice culture is a platform open to drug manipulation and real time observation of pathology in the absence of the blood brain barrier. Moreover, organotypic slices are amenable to genetic manipulation [103] and the selective ablation of cell types, such as microglia [104,105]. For these reasons, organotypic slice culture is a versatile tool for prion research.

Many different mouse-adapted scrapie strains have been shown to propagate in POSCA, including RML, ME7, 22L, 139A, and 79A, and the mouse-adapted BSE strain 301C [102,106,107,108]. Interestingly, PrP^Sc^ accumulation seems to occur on an accelerated timescale in POSCA, with RML-infected cerebellar slices having titers similar to animals at the end stage of disease by 35dpi [106]. Additionally, many pathological hallmarks of prion disease are recapitulated in POSCA, including the loss of Purkinje cells and cerebellar granular neurons by 42dpi, activation of microglia and astrocytes, and spongiform vacuolation [107]. Campeau et al., also demonstrated a loss of dendritic spines in RML-infected cerebellar slices by 63 days in culture [109]. Recently, Kondru et al., showed that cerebellar slices from Tg12 mice, which express elk PrP, could propagate CWD prions. Lysates of the cultures had seeding activity in RT-QuIC as early as 3 weeks post infection, and greater ROS were detected in infected slices compared to controls by 35 dpi [110].

Falsig et al., have demonstrated that strain features are recapitulated in POSCA. When cerebellar slices were infected with RML, 22L, and 139A prions, histoblots revealed different patterns of PrP^Sc^ deposition. Specifically, infection with 22L led to large multicentric plaques, RML to diffuse deposition throughout the slice, and 139A to a patchy pattern of deposition without white matter being affected [107]. Wolf et al., have shown how prion-infected slice cultures can be treated with GdnHCl to allow for visualization of PrP^Sc^ deposits using confocal microscopy [108]. Consequently, slice culture, in combination with confocal microscopy, may prove to be a powerful tool for investigating strain tropism. Furthermore, we have recently found that POSCA can be extended to whole brain coronal slices, which significantly increases the scope of brain regions that can be analyzed disease [101] (Figure 1).

Organotypic slice cultures provide an open, in vivo-like environment for drug screening. Moreover, less animals are needed for such experiments as many genetically identical slices can be obtained from a single mouse. Several experiments have illustrated the practicality of this model for drug screening. For instance, Falsig et al., found that pentosan polysulfate, amphotericin B, congo red, porphyrin, suramin, imatinib and E64d all had neuroprotective effects in RML-infected cerebellar slices [107]. Moreover, Cortez et al., showed that the bile acids TUDCA and UDCA had dose-dependent neuroprotective effects in RML-infected slices, with a stronger effect observed in slices treated at 14 dpi compared to 21 dpi [111]. Additionally, Kondru et al., demonstrated a reduction of seeding activity in CWD-infected slices with congo red treatment, while quinacrine was only mildly effective and astemizole was ineffective [110].

## 8. Organoids

Cerebral organoids are self-assembling three-dimensional tissues with cellular architecture that mirrors the in vivo brain. They are derived from embryoid bodies (EBs), which develop from the aggregation of either embryonic or induced pluripotent stem cells. When EBs are supplied with essential growth factors, they differentiate into specific germ lineages, such as neuroectoderm [112]. Over time, various progenitor subpopulations give rise to different types of mature neurons and glia, and these cells migrate to yield complex structures that mimic in vivo brain regions, including forebrain, hindbrain, hippocampus and choroid plexus [113]. Time and time again, compounds that show efficacy at treating prion disease in mouse models fail in clinical trials, demonstrating that mouse brains, even when they express human prion protein, are not equivalent to human brains [114]. Cerebral organoids, which are derived from human cells, get much closer to recapitulating the molecular environment of the in vivo human brain and therefore provide a more relevant system for screening drugs. Over the past few years, organoids have emerged as a promising tool to study neurodegenerative diseases including Alzheimer’s and Parkinson’s disease [115].

Recently, Groveman et al., illustrated the potential value of organoids for prion research. They produced human organoids with the MV PrP genotype and after waiting five months to allow for the development of astrocytes, the organoids were exposed to innocula from either an MV1 or MV2 sCJD case. Organoids treated with MV2 sCJD had positive seeding activity detected with RT-QuIC by 35dpi and PK-resistant PrP^Sc^ detected by western blot by 169dpi. PrP^Sc^ seemed to propagate much less efficiently in the MV1-treated organoids, with none of them showing positive seeding activity by 35dpi, and only half showing seeding activity by 169dpi. However, there was an increase in certain cytokines in MV1-treated organoids compared to control, suggesting that pathological changes occurred as a result of prion exposure [116].

Additionally, organoids offer the possibility of personalized drug screening for patients with familial prion disease mutations. Foliaki et al., prepared organoids from the fibroblasts of 2 people carrying the E200K CJD mutation, which is the most common familial prion disease mutation. Despite a line of transgenic mouse model with this mutation developing disease at about 5-6 months old [117], neither organoid showed any sign of PrP seeding activity or pathology after culturing for 12 months [118]. Again, this illustrates the difference between transgenic mice and humans, and also the need to identify methods of inducing or accelerating familial prion disease in organoids. It is also important to consider that age is the greatest risk factor for developing prion disease. Organoids, which are derived from embryoid bodies, have the epigenetics of neonatal brain tissue. Therefore, it would be useful to explore strategies for aging organoid tissue. One group have achieved more elderly epigenetics in iPSCs by exposing them to progerin, which is the mutant lamin A protein produced in the accelerated aging syndrome Hutchinson-Gilford progeria [119]. Thus, there is much room for exploration of how to best optimize iPSCs and organoids for prion research.

## 9. Enhancing Prion Propagation in Model Systems

Many techniques have been used to enhance the susceptibility of cell culture systems to prion infection and/or increase propagation of PrP^Sc^.

### 9.1. Temperature

For cell cultures to remain persistently infected, the rate at which PrP^Sc^ is being produced must be greater than the rate of cell division. Therefore, a simple method that may enhance prion propagation is to lower the temperature, which slows cell division. For instance, Taraboulos found that an immortalized line of hamster cells was able to propagate PrP^Sc^ at 34 °C, but not at 37.5 °C and hypothesized that this was due to an increase in the rate of cell division at the higher temperature [120]. Additionally, it has been shown for N2a cells that PrP^Sc^ is maintained at certain steady state levels that depend on the growth phase of the cells, and therefore their confluency [121].

### 9.2. Culture Media

The use of specific types of media and growth factors can also have a large effect on PrP^Sc^ levels. Specifically, Iwamaru et al., found that neurospheres cultured in bFGF had much more PrP^Sc^ deposition as compared with those grown in FBS [99]. Kocisko et al., found that Rov9 cells cultured in optimum produced four times the PrP^Sc^ of those cultured in MEM [20]. Bate et al., also showed that different media conditions had a large impact on PrP^Sc^ production in ScN2a cells [122].

### 9.3. PrP^C^ Expression Levels

Another factor that greatly influences the efficacy of prion infection and propagation is the expression level of PrP^C^. For instance, MG20 microglial cells derived from PrP-overexpressing Tga20 mice were able to propagate several strains of scrapie, but MG6 microglia from c56 mice, which express nine times less PrP^C^, were not susceptible to infection [42]. Similarly, MDBK cells were only able to propagate BSE prions after a lentiviral expression system was used to increase PrP^C^ expression [59]. Additionally, Falsig et al., have demonstrated that organotypic slice cultures from Tga20 mice generate much more PrP^Sc^ upon exposure to RML than slices from wildtype mice [106]. Also, a recurrent finding is that non-differentiated stem cells generally fail to sustain prion infection while their differentiated counterparts can. This may be due in part to lack of cell division and more time for PrP^Sc^ to accumulate, but differentiated cells also typically express much higher levels of PrP^C^ [60,85,97]. However, many cell culture studies involving subcloning have demonstrated that the permissiveness of prion infection in one clone vs. another is often not correlated with PrP^C^ expression levels [29,48]. Moreover, while increased PrP^C^ expression can often lead to higher levels of PrP^Sc^, it generally does not alter the range of strains that are able to propagate. Clearly, further investigation is needed to identify differences in cofactors between subclones that are differentially susceptible to prion strains.

### 9.4. Infection Techniques and Mechanisms of Cell-to-Cell Transfer

Different modes of PrP^Sc^ delivery to culture systems can have varying levels of efficacy. For example, microsomes containing PrP^Sc^ were more effective than purified fibrils for infecting both SN56 and N2a cells [34]. It is also important to consider whether PrP^Sc^ propagation occurs primarily via horizontal transmission (i.e., from cell to cell) or vertical transmission (from mother cell to daughter cells because of cell division). It has been demonstrated that N2a cells rely mainly on vertical transmission for propagation [121]. In contrast, horizontal transmission seems to occur to a greater extent in other cell lines such as HpL3-4 cells, as a greater proportion of cells become infected with each subsequent passage [40]. Techniques can be used to increase horizontal transmission between cells, such as disrupting cell membranes of infected cells [55]. Additionally, it is also necessary to consider how prions are transferred between cells, which has been shown to differ between strains [51,74]. Such transfer may occur by tunneling nanotubes, in which direct cell-to-cell contact is necessary and confluency is an important variable [123]. In other cases, exosomes may be the primary mode of transmission [124]. Simple cell culture systems are invaluable for elucidating such mechanisms of spread for different strains within a particular cell type. However, more complex systems such as organotypic slice culture and organoids can provide insight into how such mechanisms function within the more complex environment of the in vivo brain, leading strains to preferentially target specific cell types or regions.

## 10. The Influence of Strains on Different Model Systems

Not all prion model systems propagate different prion strains equally, so extrapolating results must be done with caution. Ideally one should test many strains and use model systems that are permissive to a wide array of strains. It is also vitally important to consider prion strains when testing therapeutics. A compound that works for one prion strain does not necessarily work for all, and the use of drugs can even influence the strain properties being studied. In this section, we examine considerations for the study of strains, including the changing of strain properties in model systems and drug–strain interactions.

### 10.1. Strain Adaptation

Just as successive passages of prions from one species into another leads to changes in strain properties and species adaptation, there are examples of strains becoming adapted to specific cell lines. Normally, PK1 N2a cells are refractory to infection with ME7 prions [29]. However, when Philiastides et al., exposed PK1 clones to ME7 brain homogenate, 2 out of 720 were found to propagate PrP^Sc^. After 3 more rounds of subcloning, highly ME7-suceptible PME2 subclones were detected. Intriguingly, strain properties of ME7 seemed to change upon passaging in these PME2 cells; PK1 cells that were previously refractory to ME7 could passage PME2-adapted ME7. The electrophoretic mobility of ME7 and PME2-adapted ME7 was also different. Moreover, primary cortico-hippocampal cultures from FVB mice failed to develop PrP aggregates after 2 weeks when exposed to typical ME7, but developed rod-like aggregates after exposure to PME2-adapted ME7 [31]. One explanation that has been proposed for such adaptation is the “cloud hypothesis” that states that each prion strain actually consists of many different PrP^Sc^ conformers, and that passaging in particular species or cell types selects for certain conformers more than others [125]. An experiment by Oelschlegel et al., supports this hypothesis. RML, 139A, and 79A are all mouse-adapted scrapie strains that were derived from the Chandler strain. However, depending on the system used to passage these strains, their properties can become more or less similar to each other. 139A can be distinguished from RML and 79A by its resistance to the drugs Swainsonine and Kifunensine in PK1 N2a cells. However, if 139A is first passaged in PK1 N2a cells before being propagated in mice, the resulting infectious brain homogenate becomes sensitive to these drugs, giving it more resemblance to RML and 79A [30]. A deeper understanding of the factors involved in such selection of drug-sensitive PrP^Sc^ conformers could greatly enhance the effectiveness of therapeutics.

### 10.2. Strain-Dependent Drug Effects

A recurrent lesson learned from cell culture experiments is that different prion strains can have very different responses to drugs. For instance, curcumin was shown to be a potent inhibitor of PrP^Sc^ accumulation in RML-infected N2a cells, but not in 22L-infected N2a cells [20]. Moreover, the glycosylation inhibitor Swainsonine prevents infection of PK1 N2a cells with RML but not 22L prions [126]. Hannaoui et al., demonstrated that doxycycline treatment had strain-dependent effects on MM-PrP-expressing CGN primary cultures infected with CJD agents. While treatment of cultures exposed to MM1 sCJD was effective, treatment of cultures exposed to MM2 vCJD led to a paradoxical increase in PrP^Sc^ [21]. It is therefore essential to test many different strains when screening potential anti-prion therapeutics.

Many authors have also illustrated that prions originating in different species have unique responses to drugs. Cronier et al., showed that Congo red was beneficial for reducing PrP^Sc^ propagation of hamster prion strains in CGN cultures expressing hamster PrP, but was not effective in CGN cultures infected with mouse-adapted scrapie or human CJD strains. Chlorpromazine was effective against human and hamster prions and not mouse prions, and MS-8209 was effective in all cases [86]. Kocisko et al., also demonstrated that of 32 compounds effective for treating mouse-adapted scrapie in N2a cells, only two (tannic acid and pentosan polysulfate) were effective at inhibiting PrP^Sc^ formation in sheep-scrapie-infected Rov9 cells [20]. There may be many reasons for these differences, such as the requirement of different cofactors for the replication of prions from different species. It is also possible that prions from different species activate different clearance routes and are trafficked and propagated in different cellular compartments. For instance, Ishibashi et al., found that treatment with the proteosomal inhibitor epoxomicin was beneficial in N2a cells infected with Fu-1, but was not effective in 22L or RML-infected N2a cells [127]. Evidently, further exploration is necessary to explain differences in mechanisms for the propagation and clearance of prions originating in different species.

It has also been shown that drugs have the capacity to alter strain properties and can lead to the emergence of drug-resistant prion strains. When mice were inoculated with RML and treated with the aminothiozole IND24, it seemed that a new strain, denoted RML(IND24), was generated as these mice had different lesion profiles and produced PrP^Sc^ with a different glycosylation pattern. When CAD5 cells infected with regular RML were treated with IND24, this reduced PrP^Sc^ propagation. However, treatment of RML(IND24)-infected CAD5 cells with IND24 was ineffective. Other drugs, including compound B and quinacrine also lost their efficacy [66]. Similar findings were reported by Ghaemmaghami et al., who found that treatment of RML-infected N2a cells with quinacrine led to altered strain properties and quinacrine-resistance of the resultant PrP^Sc^ [128]. One should also be aware that certain cell types seem to alter the resistance of prion strains to particular drugs. For instance, when 22L is passaged in the R33 subclone of N2a cells, the resultant PrP^Sc^ is resistant to Swainsonine. However, when 22L is passaged in the PK1 N2a subclone, the resultant PrP^Sc^ is sensitive to Swainsonine treatment and loses its ability to propagate in R33 cells [126]. For this reason, multiple cell lines should be used when testing anti-prion compounds as different cell types can have very different responses and some can lead to the emergence of drug-resistant prion strains. Future investigation should focus on the differences between cell types (such as differences in cofactors) that lead to these divergent drug responses.

## 11. In Conclusion: Choosing the Right Model

As this review has presented, there is a wealth of models available for studying prions, ranging in complexity from simple immortalized cell cultures to organoids derived from human iPSCs. The best model to use depends on the scientific question being asked. On the one hand, while cell culture lacks many of the in vivo-like features of organotypic slice culture and organoids, it is much faster, cheaper, and less technically demanding. Due to its simplicity, it is ideal for enhancing our understanding of basic biochemical aspects of prion replication and initial large-scale drug screening. On the other hand, neurospheres, organotypic slice culture, and organoids better recapitulate the cell diversity and cytoarchitecture of the human brain. Organoids and human iPSC cultures have the added advantage of providing a human genetic background. Table 4 compares the different model systems presented and highlights these considerations.

Much progress has been made over the past couple decades in characterizing prion propagation and pathogenesis in these models, and the advent of genetic engineering technologies such as CRISPR has greatly expanded the range of prion species that can be studies ex vivo. However, there is still much work to be done. Although these model systems have uncovered many potential anti-prion drug candidates, most fail to show any benefit in clinical trials. One reason for this may be the vast heterogeneity of prions; a treatment that shows efficacy against one species or strain of prion cannot be assumed effective against all prions. Thus, it is necessary to test against a wide range of strains when doing drug screening, and therefore to select a model permissive to many strains. It is also imperative that we gain further insight into the differences in cell-to-cell transfer, clearance, and pathogenic mechanisms of different prion strains. Efforts to elucidate strain-specific cofactors should also be a priority. Gaining such knowledge will greatly augment the range of therapeutic targets available and will facilitate rational drug design, bringing us a step closer to effectively treating prion disease.

## Figures and Tables

**Figure 1 biomolecules-11-00106-f001:**
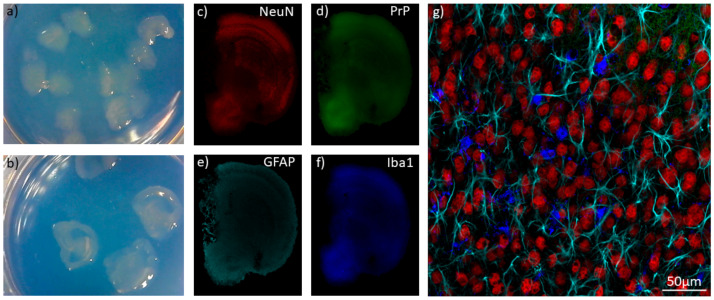
Organotypic slice culture. Cerebellar (**a**) and coronal whole-brain (**b**) organotypic slices from an 8-day-old Tga20 mouse pup; (**c**–**f**) Low magnification confocal images of a 56-day-old whole-brain organotypic slice culture, immunostained for neuronal nuclei (NeuN). (**c**) PrP; (**d**) astrocytes (GFAP); (**e**) microglia (Iba1); (**f**) demonstrating the capability of visualizing different brain regions within a slice; (**g**) higher magnification image of the cortical region, illustrating the complex in vivo-like cytoarchitecture of organotypic culture with neuronal nuclei in red (NeuN), astrocytes in cyan (GFAP), and microglia in blue (Iba1).

**Table 1 biomolecules-11-00106-t001:** Natural and species-adapted prion strains discussed in this review.

Prion Disease	Strains
Natural sheep scrapie	Kanagawa Scrapie, Obihiro Scrapie, 127S, PG127, LA404
Natural goat scrapie	At least 10 haplotypes
Mouse-adapted scrapie	Chandler, 139A, 79A, RML, 22L, 22F, 22A, ME7, 87V
Hamster-adapted scrapie	263K, 139H
Rat-adapted scrapie	139R
Bovine spongiform encephalopathy (BSE)	Typical (classic), atypical: H and L strains
Mouse-adapted BSE	301C
Transmissible mink encephalopathy (TME)	May have originated as L-type BSE
Hamster-adapted TME	Hyper (HY), Drowsy (DY)
Cervid chronic wasting disease (CWD)	Species affected: Mule Deer (MD-CWD), White-Tailed Deer (WT-CWD), Elk, Moose
Human prion disease	Sporadic Creutzfeldt–Jakob disease (sCJD): subtypes MM1, MM2, MV1, MV2, VV1, VV2 Variant CJD (vCJD) Iatrogenic (iCJD) Genetic: gCJD, Gerstmann–Sträussler–Scheinker (GSS) syndrome, Fatal familial insomnia (FFI)
Mouse-adapted CJD	SY, M1000, FU
Mouse-adapted GSS	Fukuoka-1 (Fu-1)

**Table 2 biomolecules-11-00106-t002:** Strain propagation in immortalized cell culture.

Cell Line	Cell Type	PrP Species	Strain Propagation
SMB Cells	Scrapie mouse brain cells (from a mouse that was infected with Chandler)	Mouse	Persistently infected with Chandler [22] Pentosan sulfate cured cells can propagate 22F, 139A, 79A [23] CANNOT propagate 263K [23]
PC12 Cells	Rat phaeochromocytoma cells	Rat	139A, ME7 [24,25] CANNOT propagate 263K, 139R [25]
N2a	Mouse neuroblastoma	Mouse	FU [26], Chandler, 22L, 139A [27] CANNOT propagate 87V, 22A [27]
N2a (PK1 Subclone)		Mouse	RML, 22L, 139A, 79A [28,29,30] CANNOT propagate ME7, 22A, 263K, 301C [28,29]
N2a (R33 Subclone)		Mouse	22L [28,29] CANNOT propagate ME7, RML, 22A, 263K, 301C [28,29]
N2a (PME Subclones)		Mouse	22L, RML, ME7 [31]
CAD5 Cells	Mouse catecholaminergic	Mouse	22L, RML, 139A, 79A, ME7, 301C [29,30] CANNOT propagate 263K [29]
		Hamster	263K, HY, 139H [32] CANNOT propagate DY [32]
		Bank vole	22L, MD-CWD, WT-CWD [33]
		Cervid	MD-CWD, WT-CWD [33]
SN56 Cells	Mouse cholinergic septal neuronal	Mouse	RML, 22L, ME7 [34] CANNOT propagate 87V, 263K [34]
GT1 Cells	Mouse hypothalamic	Mouse	RML [35], 22L, Chandler, FU, SY-CJD [36], Kanagawa scrapie [37] CANNOT propagate 87V, 22A [27]
CRBL Cells	Mouse cerebellum	Mouse	139A [38] CANNOT propagate RML [38]
1C11 Cells	Mouse embryonal carcinoma (neuronal stem cells)	Mouse	Chandler, 22L, Fu-1 [39] CANNOT propagate ME7, 22A [39]
HpL3-4 Cells	Mouse hippocampal	Mouse	22L [40]
CF10 Cells	Mouse neuronal	Mouse	22L [41]
MG20 Cells	Mouse microglia	Mouse	Chandler, ME7, Obihiro scrapie, BSE agent [42]
MG6 Cells	Mouse microglia	Mouse	CANNOT propagate Chandler, ME7 [42]
hTERT Microglia	Mouse microglia	Sheep	natural scrapie [43]
C8D1A Cells	Mouse astrocytes	Mouse	22L, RML [44] CANNOT propagate ME7 [44]
MovS6 and MovS2 Cells	Mouse Schwann	Sheep	Natural scrapie (VRQ allele) [45,46]
MSC-80 Cells	Mouse Schwann	Mouse	Chandler [47]
NIH/3T3 Cells	Mouse fibroblasts	Mouse	22L [48]
L929 Cells	Mouse fibroblasts	Mouse	22L, RML, 139A, 79A, ME7 [30,48] CANNOT propagate 87V, 301C [29,48]
MDB Cells	Mule deer meningeal fibroblasts	Mule deer	MD-CWD [49]
MEF Cells	Mouse embryonic fibroblasts	Bank vole	MD-CWD and WT-CWD [33]
		Cervid	MD-CWD and WT-CWD [33]
RK13 Cells	Rabbit kidney epithelial	Mouse	Fu-1, Chandler, 22L [50,51] CANNOT propagate ME7 [50], M1000 mouse-adapted CJD, MM2 mouse-adapted CJD [52]
		Bank Vole	Bank vole-adapted BSE [50] CANNOT propagate Ss3 (bank vole adapted sheep scrapie) [50]
		Sheep (VRQ allele)	natural scrapie (VRQ allele) [46,51,53,54,55] CANNOT propagate Natural scrapie (some VRQ cases and other alleles) [46]
		Goat (ARQ allele)	Goat scrapie (haplotype 1 or 2), Tg338-adapted goat scrapie (haplotype 3 and 4) [56] CANNOT propagate Goat scrapie (haplotype 3 or 4) [56]
		Elk *	Elk CWD [57] CANNOT propagate D10 Deer CWD [57]
		Human (MM)	CANNOT propagate MM2 CJD, M1000 mouse-adapted CJD, mouse-adapted MM2 CJD [52]
MDBK Cells	Madin–Darby bovine kidney cells	Bovine	Natural scrapie (VRQ and ARQ alleles) [58] CANNOT propagate BSE [58]
		Bovine **	BSE [59]
C2C12 Cells	Mouse myoblasts/myotubes	Mouse	RML, 22L, ME7 [60] CANNOT propagate Hyper strain of hamster-adapted mink encephalopathy [60]
tSP-SC Cells	Mouse stromal spleen cells (features of fibroblasts and mesenchymal cells)	Mouse	Fu-1 [61]
BMSC/336, O1BM and O2BM Cells	Adipocyte-like cells (derived from mouse bone marrow)	Mouse	Fu-1 [62,63]
HK Cells	Follicular dendritic cells.	Human (VV)	CANNOT propagate vCJD, MV sCJD, VV sCJD [64]

* Only RK13 cells expressing both elk PrP and HIV-1 GAG precursor protein (RKE-Gag cells) were able to propagate elk CWD. ** These MDBK cells were engineered to overexpress bovine PrP.

**Table 3 biomolecules-11-00106-t003:** Strain propagation in primary and induced pluripotent stem cells (iPSC)-derived cell culture.

Cell Type	PrP Species	Strain Propagation
Cerebellar granular neurons	Mouse	22L, 139A, ME7, Fu-1 [86,87,88]
	Hamster (tg7 mice)	Sc237 (Subclone of 263K) [86]
	Sheep (tg338 mice)	Natural scrapie, 139A [89]
	Human MM PrP (tg650 mice)	MM1 iCJD, MM1 sCDJ, MM vCJD [21]
	Human VV PrP (tg152 mice)	VV2 sCJD [21]
Striatal neurons	Mouse	22L, 139A [87] CANNOT propagate ME7 [87]
Cortical neurons	Mouse	22L, 139A, Chandler [87,90] CANNOT propagate ME7 [87]
Hippocampal neurons	Mouse	RML, 22L [91,92]
Primary astrocytes	Mouse	22L [88,92]
	Sheep (tg338 mice)	Natural scrapie, 139A [89]
Human iPSC-derived astrocytes	Human (MM)	MM vCJD, MM1 sCJD, VV2 sCJD [85]
	Human (MV)	MM vCJD [85]
	Human (VV)	VV2 sCJD [85] CANNOT propagate: MM1 sCJD, MM vCJD, VV1 sCJD [85]

**Table 4 biomolecules-11-00106-t004:** Pros and cons of model systems for the study of prion disease.

	Immortalized Cell Culture	Primary Culture	Stem Cell-Derived Cultures	Neuro-Spheres, Brain Aggregates	Organoids	Organotypic Slice Culture
Difficulty /Cost	Low	Medium	Medium	Medium	High	Medium-High
Time	Days to Weeks	Days to Weeks	Days to Weeks	Weeks to Months	Months to Years	Weeks to Months
Genetics	Often genetically unstable Human cell lines available	Broad choice of transgenic animals	Human genetics possible	Broad choice of transgenic animals	Human genetics possible	Broad choice of transgenic animals
Regenerative capacity	Yes	Depends on cell type	Depends on cell type	Yes	No	No
Cell diversity	Low Usually one cell type	Low Usually one cell type	Low Usually one cell type	High Most neural cell types present	Medium Cells derived from neural progenitor cells become more diverse over time	High All neural cell types present
Cytoarchitecture	Low monolayer	Low monolayer	Low monolayer	Medium 3D arrangement of cells with some in vivo-like connections	Medium-High Cell populations migrate to resemble in vivo architecture over time Not all regions present	High Mature, in vivo-like cytoarchitecture
Strain permissiveness	Low	Medium	Medium	High	More investigation needed	High
Pathology	Usually none Infected GT1 cells had abnormal morphology, autophagic vacuoles, DNA fragmentation [40]	Often increased apoptosis [79,81,83]	Increased apoptosis [89]	Increased membrane permeability, astrocyte activation [90] Loss of dendritic spines [88]	Increased neuroinflammation (cytokine release) [109]	Neuronal loss, activation of microglia and astrocytes, spongiform vacuolation [100] Loss of dendritic spines [102]

## Data Availability

No new data were created or analyzed in this study. Data sharing is not applicable to this article.

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
