# Peer review of "From Cell Culture to Organoids-Model Systems for Investigating Prion Strain Characteristics"

_biomolecules, 2021, doi:10.3390/biom11010106_

Round 1

Reviewer 1 Report

Revision biomolecules-1050512

Dear Editor,

The manuscript by Pineau and Sim is a timely and interesting review about model systems for investigating prion strain characteristics. The thematic is of interest, the authors refer to relevant bibliography and bring novel information to the field. The tables are informative, as well.

I have only minor suggestions, as you will find below:

I suggest including more information in the introduction about prion strains, as they are one of the main topics of this review. The authors comment a lot about the different model systems, but there is scarce information about each strain. I strongly suggest including a table specifying the strains that are addressed in this manuscript.

Introduction, line 28: ‘TSEs, PrP adopts a cytotoxic beta sheet-rich conformation…’ I believe that not all PrPSc forms are cytotoxic, please rephrase.

Tables 1 and 2: there is reference about disease forms and polymorphisms that are not previously cited in full in the text (or in the table legend), such as: MD-CWD, WT-CWD, MM2 CJD, VV sCJD, MM vCJD, MM1 sCJD, VV2 sCJD.

Lines 224-225: What is half of a split Gaussia luciferase? I found this sentence confusing.

Lines 306-307: I suggest including an original reference when you comment about the ola mouse.

Lines 360 and 363: please write CGN in full the first time it is cited.

Lines 509-510: ‘whole brain coronal slices, which significantly increases the scope of brain regions that can be analyzed disease [93]’ I believe this sentence is truncated.

Typos

Line 168: there is a misspelling, correct to: microglia

Author Response

We thank the reviewer for their helpful comments on our review. We have addressed all the comments as follows:

I suggest including more information in the introduction about prion strains, as they are one of the main topics of this review. The authors comment a lot about the different model systems, but there is scarce information about each strain. I strongly suggest including a table specifying the strains that are addressed in this manuscript.

We have added three paragraphs to expand this concept in the introduction and include a table of the strains discussed in the article, organized by the "parent" or natural form of the prion disease, and a listing of the strains adapted into different species hosts:

There are many different model systems in use for investigating the mechanisms of prion disease and screening drugs, from simple cell culture to organoids. However, many of the drugs shown to be effective in cell culture fail to show efficacy in vivo or in human patients [14]. This is likely due in part to the vast heterogeneity of prions, which exist as different strains. Strains are defined clinically by differences in incubation period and lesion profiles, and are thought to arise from different conformations of PrPSc [15,16]. Different strains can also be characterized biochemically by variation in PrPSc glycosylation pattern, electrophoretic mobility following PK-digestion, and conformational stability [17].

In humans, sCJD exists as six different strains that are classified based on the Prnp genotype at codon 129 (either methionine or valine), and the molecular weight of the protease-resistant core of PrPSc (21kDa in type 1 or 19kDa in type 2). These six strains are therefore denoted MM1, MM2, MV1, MV2, VV1, and VV2, with each having unique disease characteristics (Parchi et al., 1996).

Additionally, several strains of prion disease have emerged in a laboratory setting after the serial passaging of scrapie, BSE, or CJD isolates in laboratory animals of particular genetic backgrounds. Typically, the incubation period becomes shorter with subsequent passages and disease characteristics drift and stabilize, giving rise to mouse, hamster, or rat-adapted strains (Igel-Egalon et al., 2017). See table 1 for an overview of the different species and strains of prion disease that will be covered in this review.  

Introduction, line 28: ‘TSEs, PrP adopts a cytotoxic beta sheet-rich conformation…’ I believe that not all PrPSc forms are cytotoxic, please rephrase.

Rephrased: PrP adopts a beta sheet-rich conformation (denoted PrPSc) and forms fibrils and aggregates that are often cytotoxic [5,6]. Because PrPSc templates the conversion of PrPC to this aberrant state...

Tables 1 and 2: there is reference about disease forms and polymorphisms that are not previously cited in full in the text (or in the table legend), such as: MD-CWD, WT-CWD, MM2 CJD, VV sCJD, MM vCJD, MM1 sCJD, VV2 sCJD.

These have now been defined in the initial paragraphs and the new table.

Lines 224-225: What is half of a split Gaussia luciferase? I found this sentence confusing.

Clarified: ...express mouse PrP fused with either the N or C terminal half of a Gaussia luciferase. When PrP dimerizes in these cells, a luminescent signal is produced as the two halves of the split Gaussia luciferase come together. 

Lines 306-307: I suggest including an original reference when you comment about the ola mouse.

Done: Like the related SJL mice, Ola mice are known to develop tumors of the lymph nodes, Peyer’s patches and spleen in old age [Ponzio et al., 1986; Cervenakova et al., 2006]

Lines 360 and 363: please write CGN in full the first time it is cited.

Done.

Lines 509-510: ‘whole brain coronal slices, which significantly increases the scope of brain regions that can be analyzed disease [93]’ I believe this sentence is truncated.

The word "disease" was removed to make it a proper sentence.

Typos

Line 168: there is a misspelling, correct to: microglia

Corrected.

Reviewer 2 Report

This review nicely recapitulates the cellular model systems that can be used to study PrP and in general prion diseases. I think it can result useful for the scientists approaching this kind of experimental field. I recommend acceptance in the present form.

Author Response

We thank the reviewer for their positive review.

Reviewer 3 Report

Dear staff of this journal,

Thank you for submitting this manuscript

This manuscript was described as Organoids and iPS cell of prion disease and the character of iPS cell was different by Prion Strain Characteristics .

This manuscript was very excellent, the number of treatise references is also sufficient.

Author Response

We thank the reviewer for the positive review. I note that "extensive editing of English language and style required" was selected, but no specific comments were made to that effect, so we trust that this was an error in selecting the tick box.